# Pursuing Development behind Heterogeneous Ideologies: Review of Six Evolving Themes and Narratives of Rural Planning in China

**Tian Tian \*** and **Stijn Speelman**

Department of Agricultural Economics, Faculty of Bioscience Engineering, Ghent University,
9000 Ghent, Belgium; stijn.speelman@ugent.be
\* Correspondence: titian.tian@ugent.be

**Abstract:** Rural planning is in a state of flux, covering a range of topics. The objectives of planning have evolved over the years. To get an overview of the evolving themes and narratives on rural planning in China, a literature review is conducted here using text mining considering 145 papers published in Web of Science. Attention is given to trends over time in terms of the topics covered. Six evolving themes are revealed, namely: providing affordable and decent life under industrialization and urbanization progress, national ecological programs and practices, building a new (socialist) countryside and rural−urban relationship in planning, land planning and restructuring, rural tourism planning and activities, and other themes. It is highlighted that strategies and knowledge of "development" are a common instructional epistemology among agro-industrialism, agro-ruralism, scientific rationalism, and "economy oriented" humanism.

**Keywords:** rural planning; China; development; statism; neoliberalism

## 1. Introduction: Rural Planning in a State of Flux

Rural planning is a broad academic term and human practice covering a range of topics, such as rural landscape, industry development, livelihoods of villagers and farmers, environmental conservation, and health care delivery. The objective of rural planning is to achieve rural development through the allocation and management of resources, mediated by developmentalist configuration [1] (p. 25) and local communities. Rural planning could be organized at different levels, from global, national, and regional plans to plans at a village level [2] (p. 7). The time span of a rural planning is also very diverse, ranging from years to decades.

Through rural planning, social groups cast their ideas about development onto the rural space (industries, residents, and landscape). Therefore, at one time, the rural space could be described as pastoral and convivial, and at another time, it is termed as backward and peripheral. The rural space could also be seen through different lenses, such as capitalism or socialism, modernism or traditionalism, and productivism or consumerism.

Historically, it was only from the 20th century that we could truly speak of a nuanced and explicit state-sponsored rural planning and policy [3]. Ever since then, rural planning has extensively and quickly sprawled with development theories.

Since the 1930s, when a group of reformers labelled "Regionalists", around Howard Odum, sought to "fix the problems and backwardness of rural areas" on a regional basis and thus controlled local industrialization in America [3–5], agro-industrial rural development strategies have been the mainstream both in developed countries and developing countries. Two strategies are the "agriculture for industrialization" strategy embodied in the Lewis two-sector model [6], and the "industrialize the agriculture" strategy, which is associated with the continued efforts of producers and manufactures to reduce and/or regularize the importance of nature in the food production process [7]. These agro-industrial rural



development strategies are devoted to providing physical infrastructure and financial institutions, and invest in human capital, technical innovation, and social cooperatives to improve production efficiency.

In the 1930s, supply-management programs and environmental conservation were put on the agenda in North America and Europe, which was a turning point in rural planning policies. It has been widely identified that rural development should surpass the agro-industrial strategies, including the "agriculture for industrialization" strategy and the "industrialize the agriculture" strategy [7]. The agro-ruralist developmental strategies were revived again when modern states recovered from the Second World War and after the economic depression in the 1980s [3]. Attractive landscapes and leisure amenities needed to be considered under the multi-functional agriculture theory perspective and sustainable rural development paradigm. Farmers were now supposed to have dual identities: agricultural producers and stewards of the landscape. Process approaches [8] in rural planning were deployed with different stakeholders, underlining the empowerment of marginalized social groups.

Theories of rural planning and expectations of rural areas are variable, and evolve over time, and similarities and differences exist among different regions. Ellis and Biggs concluded that a first "paradigm shift" occurred in the 1960s, when small-farm agriculture switched to being considered the engine of development. The second "paradigm shift" occurred during the 1980s and 1990s, moving from the top-down or "blueprint" approach, in which rural actors were not actively involved, to the bottom-up, grassroots, or "process" approaches [8]. Frouws revealed three contested rural discourses in the Netherlands—the agri-ruralist discourse, the utilitarian discourse, and the hedonist discourse [9]. Marsden suggested three models—the mutable agro-industrial model, the bureaucratic "hygienic" model, and the relativist model—which obscure and constrain the agroecological and ecological modernization framework from taking hold, which is a more effective rural development dynamic [7]. Among these models, there is an "ascendance of certain aesthetic representations of the countryside over previous economic ones" [10], mostly because rural areas in the post-agrarian era are considered an amenity that provides the aesthetic experience of being closer to nature or a taste of rural idyll [11].

Among these enormous and complicated arguments, rural planning only has one constant attribute: it is always in a state of flux [2] (p. 9). The objectives of rural planning have evolved over the years and have broadened away from agricultural issues. In this paper, we want to give an overview of this flux in China through a review of papers published in Web of Science.

## 2. Research Methods: Text Mining

Text mining is a good method to quantify meaningful information in multiple documents. It has been popularly applied to literature reviews in biomedical and biological information research [12,13], online marketing research [14,15], and social networks and social media [16,17]. However, text mining is still rare in the scope of rural development studies. We would like to utilize this innovative method to gain insight into rural planning literature in China.

Rural planning processes and practices are supported and reflected by policy documents, laws, decrees, etc., shaped by different stakeholders. However, this research does not review the planning processes or practices, as such, but rather describes how the planning is conceived and described in the academic literature. When we started our research, CNKI (a Chinese database) was also used to collect 9575 Chinese documents about rural planning. As a result of practical problems with recognizing Chinese text in the text mining procedures, we could only use the Web of Science database. However, to get a more holistic picture on the discourse on rural planning in China and to see if there were differences between these databases, it would be interesting to do some follow-up research.

For this study, in the first step, articles were searched from Web of Science. The following search terms were used to find articles on rural planning in China: rural planning,

rural development, rural program, rural development intervention, spatial planning, rural restructuring, rural landscape, and rural construction. Details about the retrieved content are revealed in the Appendix A. This search generated 145 papers in total, after excluding unrelated and repeated articles. Cluster randomized controlled trials, social experiments, autonomous development, and institutional transformation and reform were excluded from the concept of rural planning in this paper.

In the second step, text mining was performed for all documents after excluding the references and authors' information from the publications. One measure to determine the importance of the terms in the articles is "term frequency", which calculates the frequency of a word occurring in the textual data. After tokenization of all text and lemmatization, using software R, we obtained the term frequency, as well as a wordcloud. To look at the evolving themes and narratives, tf-idf (term frequency–inverse document frequency) was considered as the second measure. The statistic tf-idf is calculated by combining the term frequency (tf), which is used to quantify how frequently a word occurs in a document, as mentioned before, with the inverse, document frequency (idf), which decreases the weight for commonly used words and increases the weight for words that are not used very much in a collection of documents. We used the tf-idf analysis to quantify how important various words were every year over all of the collected documents; in this case, the group of 145 papers as a whole. The idea of using tf-idf is to find the important terms for the content of specific years by decreasing the weight for commonly used words and increasing the weight for words that are not used very much in a corpus of gathered articles [18] (p. 35). Besides the above measures, we also extracted information on the disciplines of research and regions where fieldwork was done (panel data not included). Furthermore, a thorough review and synthesis of the literature on rural planning was also conducted. This allowed us to identify six themes on rural planning in China, which are explained in the results section.

### 3. Overview of Research on Rural Planning in China

*3.1. Overview of the Publications*

The distribution of publications per year concerning rural planning in China is shown in Figure 1. Between 1978 and 2007, the number of published papers concerning rural planning in China was very limited. Since 2008, it started to increase, and the amount has increased significantly in the past two years, reaching 32 papers in 2017. In 2018, there were already 18 publications by 24 May.

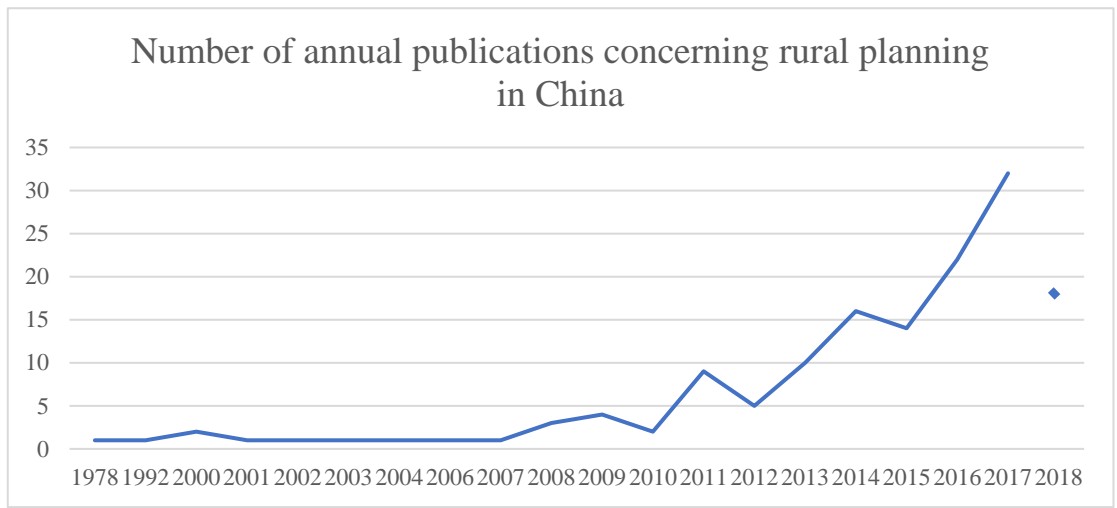

**Figure 1.** Number of annual publications concerning rural planning in China. Data last updated 24 May 2018.

In total, 14 academic disciplines were identified referring to rural planning in China. Figure 2 shows the trend in distribution across the academic disciplines. Before 2000, articles

on rural planning in China were only found in the fields of "Asian studies", "public health", and "medical sciences". Later, it was also covered in the WOS research domains of sociology, ecology, geography, anthropology, pedagogy, agricultural science, economics, management studies, engineering, architectural studies, political science, and environmental science. Since 2012, no new disciplines have been identified. The publications of the last few years were mainly found in the domains of environmental science, management studies, and geography.

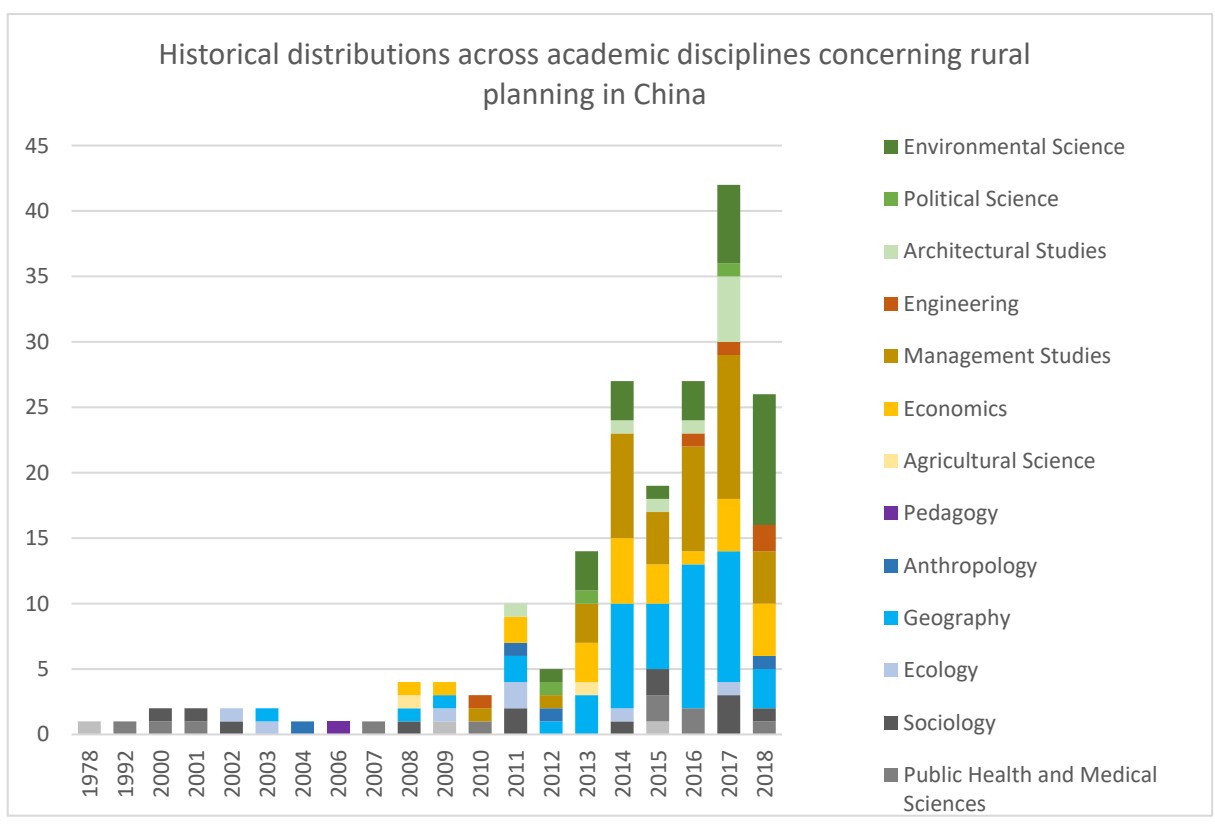

**Figure 2.** Historical distribution across academic disciplines concerning rural planning in China. Data last updated 24 May 2018.

### 3.2. Evolution in Regional Focus in Publications

Except for some publications based on national data covering the whole of China (12 articles), a number of theory reviews and framework research (12 articles), and three publications without specific locations, the bulk (115 articles) focus on regional rural planning programs in China. International comparative studies (three articles) between China and the European Union, the United States of America, and India were excluded as well. Considering the first time a specific province or autonomous region was mentioned in literature, Figure 3 shows the evolution in spatial coverage.

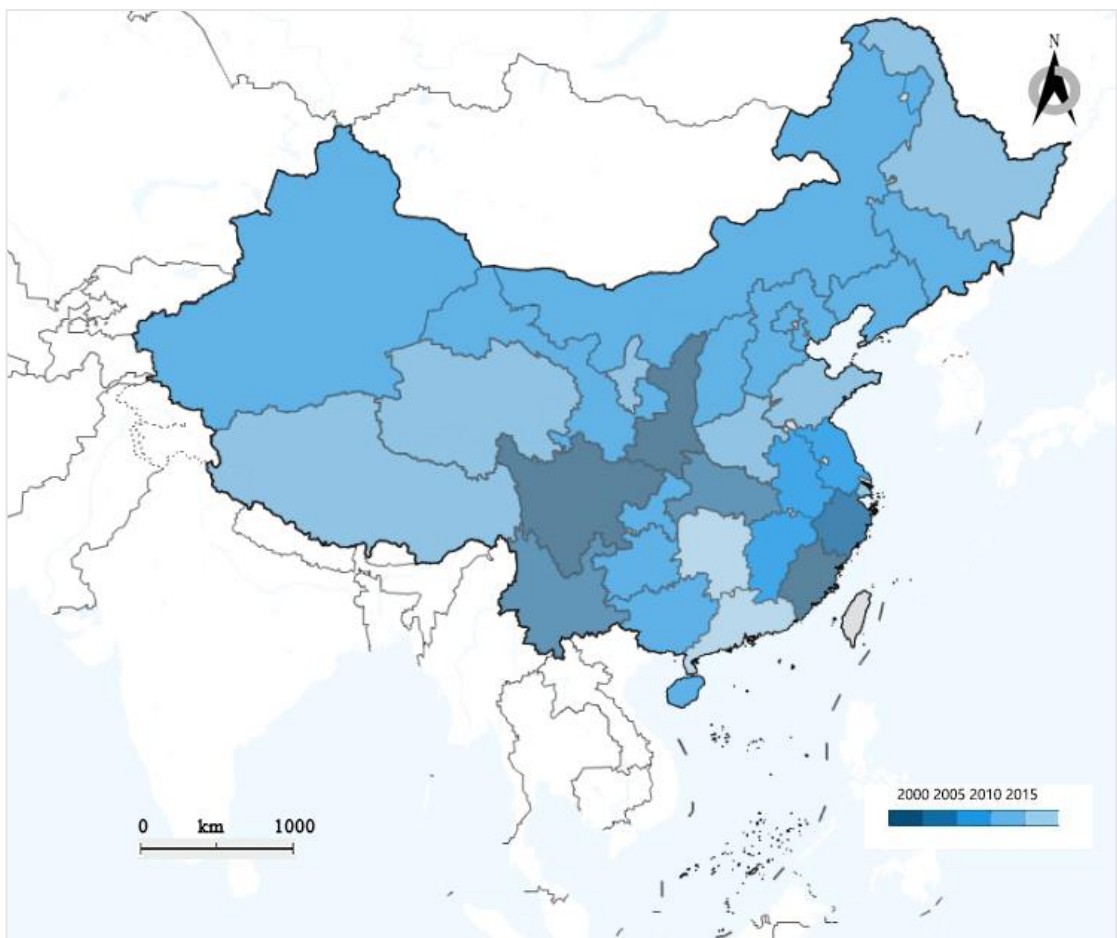

**Figure 3.** Regional distribution of publications concerning rural planning in China in different years. Data last updated 24 May 2018, no data for Taiwan.

Before 2000, only rural planning programs in Shaanxi, Sichuan, and Fujian were reported in the published papers. From 2001 to 2005, articles covering three more provinces (Yunnan, Hubei, and Zhejiang) appeared. Most provinces and regions were mentioned for the first time between 2006 and 2010. These regions spanned from the economical developed East Coast (Beijing, Tianjin, Hebei, Jiangsu, and Hainan) to the relatively laggard Central China (Shanxi, Anhui, and Jiangxi), to the economic stagnated Northeast China (Liaoning and Jilin), and finally to the underdeveloped Western China (Inner Mongolia, Gansu, Xinjiang Uygur, Chongqing, Guizhou, and Guangxi). From 2011 to 2015, rural planning in another five provinces or autonomous regions was covered in the literature for the first time. Finally, after 2015, the Hunan and Guangdong provinces were also covered. From Figure 3, neither a clear geographical pattern nor a pattern based on economic development can be derived.

### 3.3. Characteristics of Research Concerning Rural Planning in China

By applying the text mining method described in Section 2, we created a visualization of the most common words in a wordcloud (Figure 4), and compiled a table containing the top 30 most frequently appearing terms in all publications (Table 1). From the wordcloud and the table, two important characteristics were revealed: prioritizing the economy among multiple rural development focuses, and the presence of political and policy background in China.

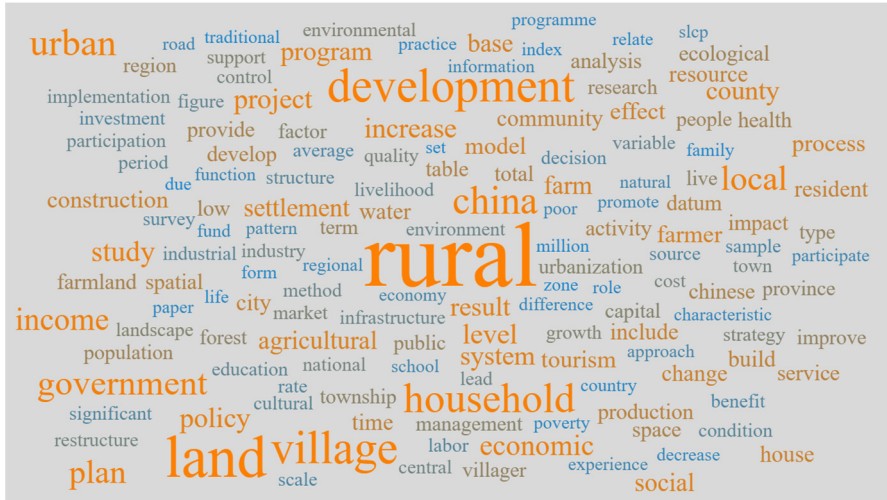

**Figure 4.** Wordcloud of the most common terms in publications concerning rural planning in China. Data last updated 24 May 2018.

**Table 1.** Top 30 frequent terms in publications concerning rural planning in China.

| Terms | rural | land | development | village | household | china | urban | government | local | plan |
|---|---|---|---|---|---|---|---|---|---|---|
| Frequency | 10,734 | 6115 | 4553 | 4475 | 3709 | 3433 | 3321 | 2843 | 2814 | 2602 |
| Terms | income | study | economic | project | level | policy | increase | program | result | system |
| Frequency | 2491 | 2109 | 2103 | 1963 | 1933 | 1931 | 1919 | 1790 | 1725 | 1673 |
| Terms | county | social | settlement | agricultural | farm | base | farmer | model | tourism | process |
| Frequency | 1660 | 1640 | 1588 | 1564 | 1513 | 1483 | 1447 | 1444 | 1418 | 1395 |

Data last updated: 24 May 2018.

(1)  Prioritizing economy among multiple rural development focuses

Rural planning in China pays attention to the economic, social, and ecological benefits of rural development. The terms "economic", "social", "ecological" and "environment" are all within the top 100 of most frequently used terms. However, other terms that are related to economic development seem to appear more often, these include "income", "production", "market", "capital", "industry", and "investment". *This suggests that the published papers have paid more attention to the economic perspective compared to social and ecological development.* How to reconcile the relation between economy, society and environment still needs to be addressed to achieving sustainable rural development.

(2)  Presence of political and policy background

The political and policy background are always present in the narratives about rural planning and rural development in China. "Government" is the eighth most frequently used word, and "policy" is on the 16th place. The words "Province", "county", "township", and "national", representing the different governance levels rank top 100 as well. Linked with macro-political narratives, we could also identify terms like "local", "household", "farmer", "villager", "resident", "community" in the list of top 100. Two other terms "process" (top 30) and "participation" (top 100) are representative for the process approach in rural planning. In this approach it is emphasized that planning should be about the process of planning rather than the production of a document (the plan) [2] (p. 9). It is furthermore characterized by the bottom-up pathway to community empowerment, in contrast with the top-down approach.

There were eight articles with the relationship between state and local actors as their research topic. A typical prejudice found in rural planning was the state−society dichotomy. National activities in rural planning are pictured as powerful and irrational, while community-based initiatives are depicted as being full of resistance spirit and creative

ideas [19,20]. The latter initiatives are ethically preferred. Since the 1980s, a growing disenchantment with the performance of state rural development agencies has been an important shared agenda across rural development actors [8], which resulted from much arm-twisting by the external financial institutions, for example, the World Bank and the UNDP. Research should pay close attention to the frequent and different interactions between state and communities in rural planning.

## 4. Six Evolving Themes and Narratives of Rural Planning in China

The measure tf-idf allowed us to find terms that are characteristic for a particular year (or several years) over all publications. We applied a text mining approach determining tf-idf on a yearly basis, starting in 2008. Because of the limited number of papers published before 2008, the publications between 1978 and 2007 were divided into three groups (Figure 5). After a thorough review and synthesis of the literature, considering the terms in Figure 5, six themes were revealed.

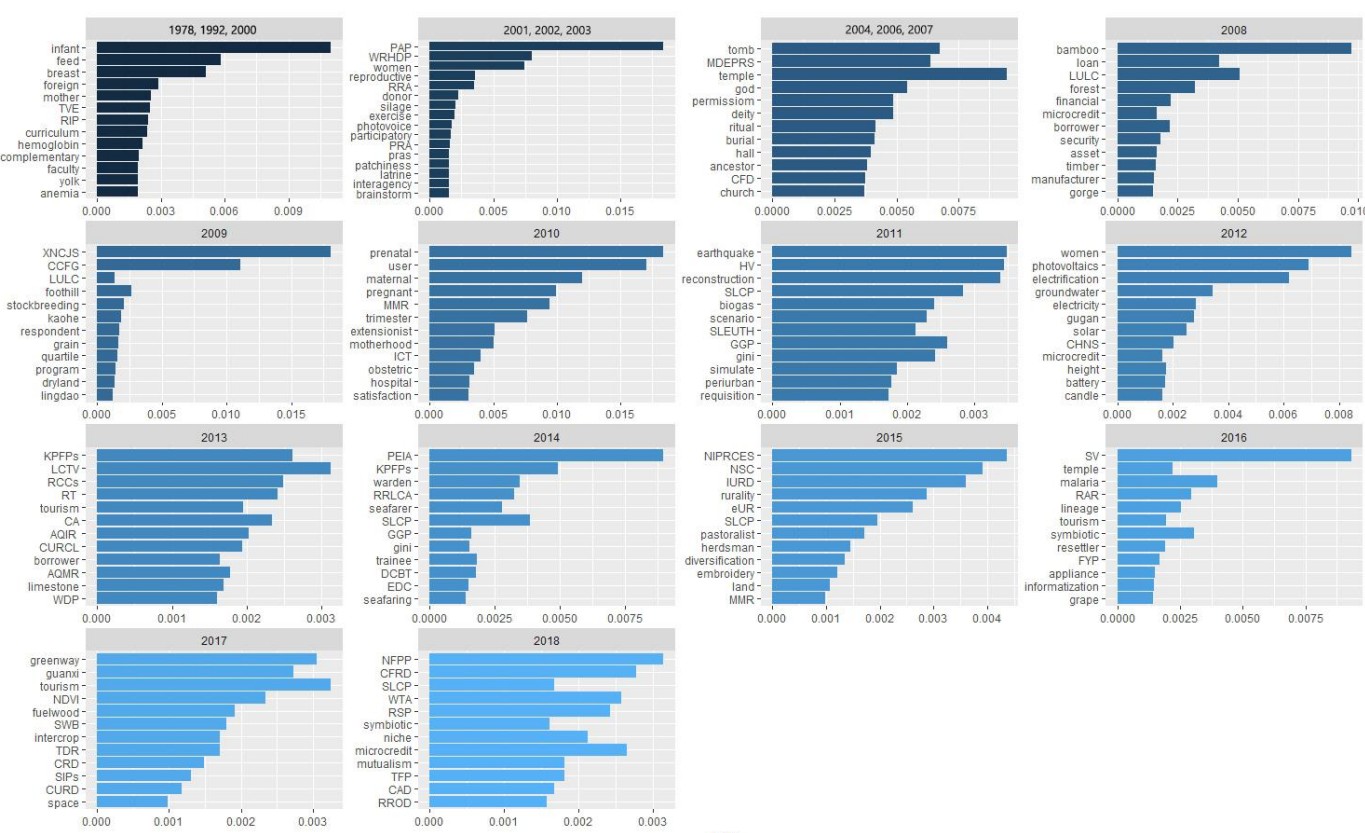

**Figure 5.** Terms with the highest tf-idf in annual publications concerning rural planning in China. Data last updated 24 May 2018.

In the last decade, publications concerning rural planning in China have started to increase covering diverse themes, which could be classified along six main topics: providing affordable and decent life under industrialization and urbanization progress; national ecological programs and practices, building a new (socialist) countryside and rural−urban relationship in planning; land planning and restructuring; rural tourism planning and activities; and other themes, including women in rural planning, cultural space, and PRA. Figure 6 shows the distribution trend of the six themes. Academic research on rural planning in China was originally only focused on the theme of providing affordable and decent life under industrialization and urbanization progress. Since 2009, the themes regarding national ecological programs and building a new (socialist) countryside have become prevalent. They were followed by the theme of land planning and restructuring, which has

become the most dominant research topic recently. Finally, rural tourism appeared as a new subject in the last years.

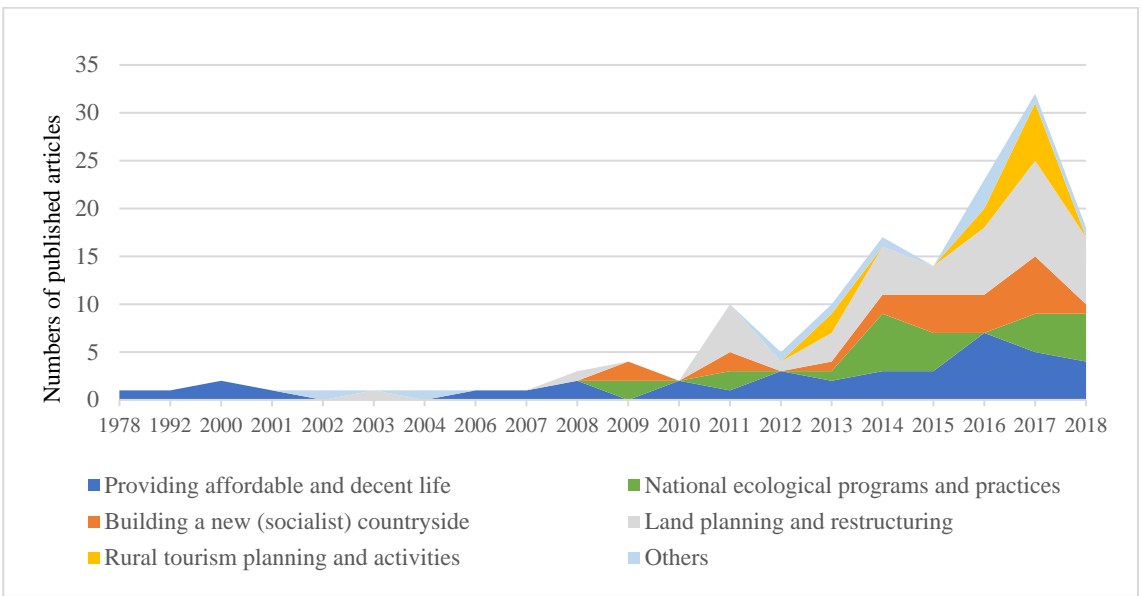

**Figure 6.** Historical distribution of evolving themes concerning rural planning in China. Data last updated 24 May 2018.

### 4.1. Providing Affordable and Decent Life during Industrialization and Urbanization Progress

Academic research on rural planning in China focuses on rural industries (with key terms as "Township and village enterprise (TVE)" and "Rural Industries Projects (RIP) in 1978, 1992, and 2000; and "timber" and "manufacturer" in 2008; "seafarer" in 2014, Figure 5), on rural health services (with "Women's Reproductive Health and Development Program (WRHDP)" in 2001, 2002, and 2003, and "hospital" in 2010), on education programs (with "curriculum" in 1978, 1992, and 2000, and "The Modern Distance Education Project for the Rural Schools (MDEPRS)" in 2004, 2006, and 2007), service infrastructure ("latrine" in 2001, 2002, and 2003, and "electricity" and "groundwater" in 2012), and on rural financial institutions ("loan" and "financial" in 2008; "borrower" in 2008 and 2013; and "microcredit" in 2008 and 2018). Before 2008, the dominating theme in rural planning was to provide an affordable and decent life under industrialization and urbanization. This is considered among the responsibilities of the state to retain its' legitimacy [21]. This theme is not likely to vanish, even though it is not in the priority position anymore.

Providing public goods (infrastructure, health services, education, and financial institutions, etc.) and promoting the development of rural industries and living standards, belong to a part of state-building, even though there are a lot of arguments on the sequencing of state-building [22]. As Lapping said, all of these were to be done through a national commitment to social planning [3] implemented not only on a regional basis and for locally controlled industrialization, but also on the state demand and global neoliberalism.

### 4.2. National Ecological Programs and Practices

Since 2009, national ecological programs emerged as an important theme in publications. As we can see in Figure 5, "Conversion of Cropland to Forest and Grassland Program (CCFG)" and "grain" in 2009; "Sloping Land Conversion Program (SLCP)" in 2011, 2014, and 2015; "the Grain for Green Program (GGP)" in 2011; and the "Key Priority Forestry Programs (KPFPs)" in 2013 and 2014, all are important terms in the corresponding years.

The Grain for Green Program (GGP) attracted considerable attention. This program is sometimes also termed the Sloping Land Conversion Program (SLCP) or the Conversion of Cropland to Forest and Grassland Program (CCFG). It intends to convert cultivated land on steep slopes (over 25° slops, and 15–25° cropland close to important water sources)

to forest and grassland by compensating participating households with a cash subsidy, free seeds and seedlings, and in-kind grain allocations. Since the beginning of the 21st century, China has launched several national ecological programs, which are collectively named the Priority Forest Programs by scholars. They are designed with the goal to improve the ecological environment, to alleviate poverty, and to adjust the industrial structure. However, more than half (12 out of 20) of the publications mentioning the national ecological programs only focused on the impact on rural household incomes and livelihoods, and four papers were interested in the households' attitude, and another four articles included ecological, economic, and social perspectives together. Not until 2017 were four articles published that concerned the ecological environment investigation, ecological inter-correlation, and ecological construction in rural areas.

To analyze human–nature relationships in rural planning, the agroecological perspective implied a concept of "co-evolution". Sustainable rural development containing ecological, economic, social, and productive development together was put forward. However, the ecological environment is often still translated in the economic dimension, and a process of institutionalization of "ecology" re-embed it into economic practices. For example, payments for ecosystem services became an important policy mechanism. They are economic instruments that are trying to create or change stakeholders' incentives and behavior to promote ecological restoration and/or conservation [23]. Research has shown that economic and financial concerns are still the most important drivers for rural households in ecological programs [24].

### 4.3. Building a New (Socialist) Countryside and Rural−Urban Relationship in Planning

The ecological programs funded by the government in China, which appear to be campaign-style and top-down programs, steered by government decision-making mechanisms inherited from central planning policies [25,26], have also appeared been applied in rural restructuring programs in a "blueprint" style. The political slogan "building a new (socialist) countryside" was promulgated by China's National People's Congress in 2006. Publications focusing on this topic arose since 2009, making "build a new socialist countryside (shehui zhuyi xin nongcun jianshe, XNCJS)" or "new socialist countryside (NSC)" key terms in in 2009 and 2015 (Figure 5).

"Building a new (socialist) countryside" is the national rural planning goal in China, which is aimed at prompting local governments to reorganize, streamline, and focus their efforts to promote comprehensive rural development. It primarily focuses on infrastructural and agricultural modernization linked to ecological sustainability, and on the provision of public goods [27]. An interesting point found in our research was that international scholars used the term "building a new socialist countryside", while Chinese scholars always used "building a new countryside" without the attribute "socialist". Some research identified "building a new socialist countryside" as a "managed campaign" [28] or "top-down campaign" [29], because it powerfully combined technocratic pragmatism with Mao-era campaign tactics, such as work teams (named as lingdao xiaozu in Chinese, as shown in Figure 5) [28]. Some studies suggest it was not just an empty slogan, but a macro-policy with meaningful implications, which manifested a development model as the historical experiences of Europe and industrialized East Asia [27].

For the rural−urban relationship in planning, integrated/coordinated urban−rural development (as IURD and CURD in Figure 5) has become a consensus among scholars [30–32] and policymakers. However, institutional obstacles between urban and rural areas are enormous in China, and the same holds for the gap in resource allocation. Rural planning underperformed [33] compared with urban planning. Rurality is presented as backward and less-developed in some research [34], and the way to achieve integrated/coordinated urban−rural development is simply to abruptly industrialize and modernize villages. On the contrary, some planners and governments sought to avoid replicating urban settlements in rural areas by developing recognizably "pastoral" villages with environmental amenity and a stock of arable land to "make the countryside

more like the countryside" [35]. In China, there is still a long way to go to reach integrated/coordinated urban−rural development, and the continuous of rural practices creates a diverse experience [36].

### 4.4. Land Planning and Restructuring

Land is an important topic in China, and we can see from Figure 5 and Table 1 that "land" is the second most frequently used term in the publications. The theme of land planning and restructuring in rural China became widespread since 2011, after it was first introduced in publications in the period 2003 to 2008. In China, explosive urbanization and burgeoning non-agricultural industries have increased pressure on rural land. There are several articles focusing on the spatiotemporal evolution of land-use and land-cover change [37–41]. They all found that rural settlements and farmland declined accompanied with the sprawl of urban areas.

Facing challenges in protecting farmland, the Ministry of Land and Resources announced an obligatory target in 2006 that farmland must remain above 1.8 billion mu (120 million hectares) in China. Meanwhile, a land-use policy, termed "Linking the Increase in Urban Construction Land with the Decrease in Rural Construction Land", was introduced to achieve equilibrium in the supply of land by balancing urban construction land and rural construction land. If rural construction land is to be reclaimed as arable land, the quota of land for industrial growth and urbanization will increase. It is against this backdrop that rural settlement land planning and restructuring are achieved through centralized residential districts [42] and resettlement for villagers [43].

There was found to be a widening gap between the prices of urban construction land and rural construction land. State-led land requisition was dominant, which is decided by dual-track land ownership [44] in China. This institutional foundation makes it possible for the state to be involved in space commodification. Enormous political and economic profits are the impetus of the state and market, behind the pursuit of land equilibrium and "integrated/coordinated urban−rural development". The interweaving between different levels of governments and external investments was reported in the research [42–45]. Meanwhile, the Chinese state's fragmented authority provides a favorable institutional environment for a committee of villages to bring their own land directly to the land market and to reap large profits [46]. Self-organized rural planning, democratic decision-making, and endogenous institutional innovation [47] have been reported as well.

The complex interplay between different stakeholders, especially the state and market, has led to the multi-faceted transformation of rural communities and to a complicated countryside profile [48].

### 4.5. Rural Tourism Planning and Activities

Rural tourism planning and activities were present in China through poverty alleviation programs in 2013, as shown in Figure 5, and "rural tourism (RT)" and "tourism" arose a with high tf-idf in 2013, 2016, and 2017 (Figure 5). Rural tourism started off as one of the solutions for rural socio-economic restructuring and poverty alleviation since the early 1980s [49], dominated by China National Tourism Administration and different levels of government. Several articles underlined the importance of integrated rural tourism, such as the "LCTV framework" [50] (Figure 5). Rural tourism has often focused on economic priorities, with little concern for carrying capacities or the negative impacts on local communities [51], especially ecological and socio-cultural impacts.

Under bureaucratic top-down political command structures and business−bureaucracy partnerships, different levels of government, external tourism enterprises, and state-owned tourism enterprises have occupied important positions. Whether and how local communities participate in and benefit from rural tourism has been a concern [52–55]. In Keyim's research, the local community did not have many socio-economic benefits as a result of its unequal relationship with the other involved actors and their poor participation in the local tourism development processes [52]. However, in the research of Chen and Liu et al.,

the indigenous residents were the primary actors engaged in tourism through informal institutions, while small business owners who are non-local residents were excluded [54,55]. Even though local communities could benefit, their participation in the formal decision-making process usually is very limited. Even if a rural community was involved in public participation activities, institutional shortfalls existed [53].

### 4.6. Other Themes: Women in Rural Planning, Cultural Space, and PRA

Women appeared in rural planning literature in the role of "mother" in 2000. Until 2010, women were always connected with "reproduction", "caregiver", and "motherhood" in rural development programs focusing on rural health services and children's nursing, as shown in Figure 5. Only in 2012, an article that focused on women's participation and empowerment in rural development projects appeared [56].

Rural cultural space received researchers' attention in the perspective of anthropology, which shows the complexity and flexibility of the local people's interaction with the state in rural planning [57,58]. The promotion of the replacement of earth burials with cremation; converting ancestor temples into museums, cultural halls, or elderly activity centers; registering shrines, temples, and churches; and even campaigns to dismantle temples and churches occurred in China recently. The conversion of traditional and lineage-based spaces into modern and authorized cultural spaces is not only from the demand of reinforcing the authority of the government in rural areas, but is also accompanied with capital interests of land.

Participatory Assessment and Planning (PAP) and Participatory rural appraisal (PRA) approaches appeared in the literature concerning rural planning in China as of 2002 (Figure 5). In case studies, PAP and PRA approaches were used to formulate action plans of rural development [58] and to conduct evaluations for rural development interventions [59]. These approaches enabled all relevant actors to be involved in a "doing and learning" process, in which the community could generate new knowledge, acquire new skills, build its capacity and confidence to interact with others, and innovate new approaches and technologies for its development [59]. This envisages rural planning as a participatory process that empowers rural dwellers to take control of their own priorities for change [8], along with emphasizing the uniqueness of local knowledge and individual experience, while rejecting overarching theories and large-scale programs.

### 5. Conclusions and Discussion: The Pursuit of Development behind Heterogeneous Ideologies

Research and practices on rural planning in China are heterogeneous. Both agro-industrial and agro-ruralist ideologies exist at the same time. On the one hand, "building a modernized countryside" aimed to provide an affordable and decent life under industrialization and urbanization progress; on the other hand, "building the countryside more like the countryside" yielded a lot of action, especially in the theme of rural tourism, which underlined a highly nostalgic and romanticized view of rural life. This paradox co-exists especially in the theme focused on land planning and restructuring and building a new (socialist) countryside. Some planners and governments endeavored to either urbanize villages or resettle villagers in urban areas [43]. Some planners sought to avoid replicating urban settlements in rural areas by developing recognizably "pastoral" villages, an approach that is being widely echoed in the relatively new discipline of rural spatial planning in China [35].

Scientific rationalism appeared together with humanism. Rural planning could be more strategic, systematic, scientific, data-based, and security-oriented, especially in geographical and ecological perspectives [34,60]. However, humanism is the epistemological premise of sociology and anthropology with a focus on rural society, culture, relationships, and interactions between social groups and society. Several studies in the field of geography also included a symbiotic system with the "people oriented" idea in rural settlement restructuring [61]. However, sometimes "people oriented" development becomes

"economy oriented" development, as we identified in the theme of national ecological programs [23,24].

There is a common instructional epistemology among agro-industrialism, agro-ruralism, scientific rationalism, and "economy oriented" humanism: development, which was the third most frequent term in all of collected publications (Table 1 and Figure 4). The consolidation of the discourse and strategy of development starts with the problematization of poverty, strengthened by principal mechanisms through which development has been deployed, namely, the professionalization of development knowledge and the institutionalization of development practices [62] (p. 17). In rural planning, economic growth, capital accumulation, technological advocating, and modernization are outstanding examples of development.

The prevalence of development is closely connected with statism and neoliberalism. The involvement of the state in rural planning constitutes one of the most conspicuous characteristics in modern China. The degree of involvement of the modern state in the rural areas also used to constitute one of the most noteworthy and original features of post-war history in Europe [63] (pp. 51–56). The financing and provision of research, vocational education, and extension work for agriculture has been left to the state, as well as providing much of the access to finance other types of input. The state is also involved in the commercialization of farm produce, attempting to regulate and organize markets and marketing. The state has additionally assumed that the government should be concerned rural welfare, such as education, medical service, endowment insurance. However, in China, the practice of statism is broader and deeper. The modern state assumes responsibility of the management of not only the rural economy, society, and ecology, but also rural family and culture, included birth giving, restructuring settlements [42,43], and burials [57].

At the same time, the state stands firm and attempts to bring as much rural affairs as possible into the domain of the market, moving China towards neoliberalization. Rural planning is inundated with commercialization, monetization, and financialization. The state helps to promote market-oriented rural industries [64], establish rural financial institutions [65], re-embed ecology into economic practices by payments for ecosystem services [66], frame policies on property rights of rural land [46], and market rural culture and rural image for urban dwellers [52].

Rural planning is about setting a common vision for rural areas. This is a tough task in China, "where the traditions have not yet left and modernity has not settled in" [62] (p. 218), where heterogeneous ideologies are present. While development-oriented rural planning is dominant in China under the impact of statism and neoliberalism, popular practices have not disappeared. Popular culture has been revived in the expanding space of homogeneity created by the modern state and global capital [57,58], and self-organized rural planning, democratic decision-making, and endogenous institutional innovation [47,67] are in progress.

It is a cliché that rural planning is not only for achieving economic development. However, the importance of social inclusion, local culture, biodiversity, and ecosystem integrity still should be recognized and put into practice in China. Therefore, it is crucial to always have interdisciplinary teams and multi-stakeholder participatory bodies in the process of planning. How to coordinate local communities, developmentalist configuration, and institutional mechanisms needs to be addressed to ensure that local interests and priorities are represented in rural planning. In the EU, the LEADER/CLLD local development approach has provided a good model that involves local partners in shaping the future development of the countryside.

**Author Contributions:** Conceptualization, T.T.; methodology, S.S.; software, T.T.; supervision, S.S.; validation, S.S.; visualization, T.T.; writing—Original Draft Preparation, T.T.; Writing—review and editing, S.S. The authors participated in the research topic and share joint responsibility for this work. All authors have read and agreed to the published version of the manuscript.

**Funding:** This research received no external funding.

**Institutional Review Board Statement:** Not applicable.

**Informed Consent Statement:** Not applicable.

**Conflicts of Interest:** The authors declare no conflict of interest.

## Appendix A

**Table A1.** Search key words and results from Web of Science.

| Searched for | Document Types | Timespan | Indexes | Number of Papers |
|---|---|---|---|---|
| TOPIC: (rural development China) AND TITLE: (rural development) | ARTICLE | All years | SCI-EXPANDED, SSCI, A&HCI. | 237 |
| TOPIC: (rural planning China) AND TITLE:(planning) | ARTICLE | All years | SCI-EXPANDED, SSCI, A&HCI. | 100 |
| TOPIC: (Rural program China) AND TITLE: (rural program) | ARTICLE | All years | SCI-EXPANDED, SSCI, A&HCI. | 82 |
| TOPIC:(Development intervention rural China) AND TITLE: (rural) | ARTICLE | All years | SCI-EXPANDED, SSCI, A&HCI. | 65 |
| TOPIC: (spatial plan rural China) AND TITLE: (rural) | ARTICLE | All years | SCI-EXPANDED, SSCI, A&HCI. | 42 |
| TOPIC: (rural restructuring China) AND TITLE:(rural) | ARTICLE | All years | SCI-EXPANDED, SSCI, A&HCI. | 64 |
| TOPIC: (rural landscape China) AND TITLE: (rural) | ARTICLE | All years | SCI-EXPANDED, SSCI, A&HCI. | 105 |
| TOPIC: (rural construction China) AND TITLE: (rural) | ARTICLE | All years | SCI-EXPANDED, SSCI, A&HCI. | 171 |

Data last updated: 24 May 2018.

**Table A2.** Abbreviations and interpretations in Figure 5.

| Abbreviations | Interpretation |
|---|---|
| TVE | Township and village enterprise |
| RIP | Rural Industries Projects |
| PAP | Participatory assessment and planning |
| WRHDP | Women's Reproductive Health and Development Program |
| RRA | Rapid rural appraisal |
| PRA | Participatory rural appraisal |
| MDEPRS | Modern Distance Education Project for the Rural Schools |
| CFD | Care for development |
| LULC | Land-use and land-cover |
| XNCJS | Build a new socialist countryside (shehui zhuyi xin nongcun jianshe) |
| CCFG | Conversion of Cropland to Forest and Grassland Program |
| Kaohe | Evaluation |
| Lingdao | Leading |
| MMR | Maternal mortality rate |
| ICT | Information and communication technology |
| HV | Hollow Village |
| SLCP | Sloping Land Conversion Program |
| SLEUTH | SLEUTH is a self-modifying probabilistic cellular automata (CA) model; the model's acronym is an abbreviation for the initials of input data layers: slope, land use, exclusion, urban, transportation, and hill shade |
| GGP | Grain for Green Program |
| CHNS | China Health and Nutrition Survey |
| KPFPs | Key Priority Forestry Programs |

**Table A2.** *Cont.*

| Abbreviations | Interpretation |
| --- | --- |
| **LCTV** | LCTV is a framework created from landscape management, community acceptance, tourism activity, and visitor satisfaction |
| **RCCs** | Rural credit cooperatives |
| **RT** | Rural tourism |
| **CA** | Cellular Automata |
| **AQIR** | Average quintile immobility rate |
| **CURCL** | Coordinating Urban and Rural Construction Land policy |
| **AQMR** | Average quintiles move rate, reflecting the income mobility of rural households |
| **WDP** | Western Development Program |
| **PEIA** | Plan environmental impact assessment |
| **RRLCA** | Rural residential land consolidation and allocation |
| **EDC** | Despite this professional and modernist transformation, the formation of a shareholding company, known as the Economic Development Company (EDC), did not fundamentally change the nature of the collective system |
| **NIPRCEs** | Nutrition Improvement Program for Rural Compulsory Education Students |
| **NSC** | New socialist countryside |
| **IURD** | Integrated urban−rural development |
| **eUR** | Urban−rural balance: efficiency and equity |
| **SV** | Specialized village |
| **RAR** | Rural appliance rebate |
| **FYP** | Five-year plan |
| **NDVI** | Normalized difference vegetation index |
| **SWB** | Subjective well-being |
| **TDR** | Transfer of development rights |
| **CRD** | Coordinated rural−urban development |
| **SIPs** | Structure insulated panels |
| CURD | Coordinated Urban and Rural Development |
| NFPP | Natural Forest Protection Program |
| CFRD | China Foundation for Rural Development |
| WTA | Willingness to accept |
| RSP | Relocation and Settlement Program |
| TFP | Total factor productivity |
| CAD | Characteristic agriculture development |
| **RROD** | Rural Road-Oriented Development Model |

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
