# Peer review of "Pursuing Development behind Heterogeneous Ideologies: Review of Six Evolving Themes and Narratives of Rural Planning in China"

_sustainability, doi:10.3390/su13179846_

Round 1
Reviewer 1 Report
Dear Authors,
I found your article to be of high quality and a contribution to the literature on rural planning in China.
I have a few suggestions and comments that may I hope improves your paper.
- While the title is good, it would be enhanced if there was a subtitle that expressed your main point or thesis in the paper. What is the one conclusion you want to convey after reading all these articles (145) on rural planning in China? I think if you conveyed this point in a subtitle the title would be enhanced and better inform the reading from the beginning of your main claim in the paper. The subtitle should follow the title and a colon.
- The Abstract is fine but an epigraph, or brief quotation, that captures your main point in the essay would enhance the paper You could take a quote from one of the articles you refer to in the paper.
3. Good to provide sections with section headings to your paper beginning with the Introduction.
4. P. 1, para. 1. "achieve rural development through interventions." What kind of "interventions?" by State? NGOs? Corporations? etc. The type of interventions you have in mind should be expressed here.
5. P. 1, para. 1. "The time span of a rural planning is also very diverse." From what to what is the time span you have in mind? What is "very diverse?" The time it takes to do what? should be expressed here.
6. P. 1, para. 3. "and legible state-sponsored." The word "legible" here seems misplaced since it usually refers to writing and its comprehensibility. so you mean something like: "explicit," or "distinct"?
7. P. 2, para. 2. "the Lewis two sector-model." First name of the person referred to needed here, without an apostrophe since you are merely naming the model. Also, Lewis not cited in References and should be since referred to. In addition, what this model is and does should be explained here.
8. P. 2, para. 3. The ideas in this paragraph need a source to be cited here. Also, the nature of the "agro-industrial" model needs to be explained and source provided.
9. P. 2, para. 4 "top-down or 'blueprint' approach to rural development". This approach needs to be explained.
10. P. 2, para. 4. "Aesthetics and theories of rural planning". It is surprising to see a reference to "aesthetics" in this context. This should be explained: what do you mean by "aesthetics" and why and how does it apply to rural planning?
11. P. 2, para. 4. "slightly closer to rural sustainability". The use of "slightly" here is curious: why only "slightly"closer to rural sustainability? Why not "full" rural sustainability?
12. P. 3, para. 1. "rural planning . . . is always in a state of flux." That may be true, but is it also true for practice, or implementing the ever-changing policy? It would be surprising if the implementation of policy were so changeable or be in a "state of flux." Need to consider and discus and clarify to be rationally useful.
13. P. 3, para. 3. articles about 'institutional transformation and reform were excluded from the concept of rural planning in this paper." Why? Need to explain and defend to be rationally persuasive.
14. P. 4, para. 2. and several other locations in the paper: The word "ERROR" occurs in bold regarding missing references. These missing references must be provided to make the article ready for publication. Please correct each every occasion where the citation is missing.
15. P. 7, para. 1. "This suggests that the published papers have paid more attention to the economic perspective etc." This an important conclusion and should be underlined or italicized to highlight for the reader. It could also provide the subtitle conclusion for the title. But also this conclusion should be commented upon and theorized.
16. P. 7, para. 2. "process approach in rural planning" This idea is not explained but should be; what do you mean by this "process approach?"
17. P. 7, para. 3. Claims are made about a "methodological prejudice" and about "the relation between the state and communities" in this paragraph. Both claims should be explained and defended further to be rationally persuasive.
18. Section 4, P. 1 repeated in my copy, para. 1. 6 themes are revealed from your text mining. Could be used in subtitle.
19. P. 3, Second p. 3, Section 4, para. 3. "rural construction land that are to be cleared up" etc. What does "cleared up" mean here? Should explain further.
20. P. 4, Second p. 4, para. 6, "participation and empowerment in rural development planning and programs". Need to discuss the role of democracy in rural planning among the community members and why this is a value? How to get community members mobilized to participate in planning? how to get planning professionals able to utilize ideas of the uneducated community members and how to present the planning technicalities to lay audiences should be discussed.
21. P. 5, Second P. 5, Section 5, para. 2. "aesthetic of nostalgia". What does this mean? Why it is important? Need to discuss and clarify.
22. P. 6, Second P. 6, Section 5, para. 3. "sprawl of statism". What does this mean? need to clarify and explain.
23. P. 6, Second P. 6, Section 5, para. 5. "This contradictions between sprawl of statism and neoliberalism have unified in China." The wording in this important conclusion is unclear to the reader. How can contradictions "unify" China? Must explain and defend this key conclusion of the paper.
24. P. 7, Second P. 7, Section 5, para. 2, last sentence. "Popular Chinese culture [? what is that? not discussed.] has revived in the hegemonic sociocultural space, and self-organized rural planning, democratic decision-making, and endogenous institution innovation are in progress" This is the most important conclusion, if I understand it correctly, yet how it follows from text mining needs to be discussed and supported to make the article's methodology justified. Also. what is meant the key terms like "Popular Chinese culture." "hegemonic socioculture space," and "self-organized planning" and "democratic decision-making," and "endogenous institution innovation" all need clarification, or at least references to the parts of the paper that discuss them.
I realize that referring to Harvey and not discussing his several controversial claims saves time and writing, and that is acceptable, but this final sentence seems like it is dogmatic without further clarification and defense and relating to the text mining which is the heart of the paper's argument.
I hope these comments and questions help you improve the article, at the very least the references where absent need to be provided.
Author Response
We are grateful to the reviewer for detailed comments and insightful suggestions on the manuscript. We have been able to incorporate changes to reflect all suggestions provided by the reviewer. Please see a point-by-point response in the attachment.

Reviewer 2 Report
the authors have addressed my comments and made appropriate revisions. I suggest the paper be accepted in the current form.
Author Response
We thank the Reviewer for their detailed comments and review, which helped a lot to improve the manuscript.

Reviewer 3 Report
A very interesting paper with a thoroughly conducted review.
Some general comments:
1. The authors have to change the reference style and redo pageing because it is rather confusing.
2. In general there is a very interesting discussion but I fail to see the connection with the research conducted. The discussion has to do with the actual policy formulation process while the research was a quantitative review of published research. How do the authors connect these two issues? It would be much better if the authors have selected some more of the papers (they already did that) and based on the content, inform the discussion.
Some specific remarks
|
LIne |
Text |
Comment |
|
Page 7 |
Another two terms ‘progress’ and ‘participation’ represent the process approach in rural planning, they characterise the bottom-up pathway to community empowerment, in contrast to the top-down approach |
The authors do not provide information on the ranking. |
|
Page 7 |
Research should pay close attention to the frequent and different interac-tions between state and communities. Neither is there a vacuum between the state and local communities, nor is the relation be-tween the state and communities as monolithic as people assume. |
It is very interesting statement with which I totally agree but I fail to see its relevance in the text. |
|
Page 3 (??) lines 111-119 |
To face challenges of farmland protection, the Ministry of Land and Resources introduced an obligatory target in 2006 that farmland must remain above the red line of 1.8 billion mu (120 million hectares)3. But a breach was opened in the same year by ‘Linking the Increase in Land Used for Urban Construction with the Decrease in Land Used for Rural Construction’. This was piloted in 2006 and formally adopted in 20084. Under these policies, land equilibrium is achieved by balancing urban construction land and rural con struction land that are to be cleared up and reclaimed as arable land. Rural settlement land planning and restructuring are achieved through centralized residential districts (Huang et al. 2013) and resettlement for villagers (Lo, Xue, and Wang 2016). |
This is not very clear, please clarify. |
Author Response
We are grateful to the reviewer for insightful suggestions and detailed comments on the manuscript. We have been able to incorporate changes to reflect all suggestions provided by the reviewer. Please see a point-by-point response in the attachment.

Reviewer 4 Report
Rural planning is an important part of rural sustainable development, which has always been the focus of academic and political circles. Based on the analysis method of text mining, 145 English articles on rural planning in China are systematically sorted out, and some preliminary conclusions are obtained. In general, this study has certain significance, and the method of text mining is also new to some extent. However, there are still some shortcomings that need further improvement. Specific comments are as follows:
(1) The selection of research objects. The author focuses on China's rural planning and analyzes 145 SCI/SSCI retrieved articles from the Web of Science. Personally, I think the selection of such analysis objects may be biased. Since China has a long history of thousands of years, there have been quite a lot of works on rural planning, but most of these works are written in Chinese. Therefore, to analyze the problems of rural planning in China, is it better to use Chinese text analysis?
(2) The normative problem of charts. For example, In Figure 3, the use of China's map is obviously non-standard, and it is recommended not to use this map because it lacks too many elements (such as lack of compass, lack of scale, and the south China Sea islands in the lower right corner are also non-standard).
(3) The study needs further discussion. The author has made some analysis based on the text data, so in the end, some suggestions should be given based on the actual situation (China + other countries in the world).
Author Response
We are grateful to the reviewer for insightful suggestions on the manuscript. We have been able to incorporate changes to reflect most of the suggestions provided by the reviewer. Please see a point-by-point response in the attachment.

Round 2
Reviewer 3 Report
I would like to congratulate the authors for the improvements.
Unfortunately, the reference style [Numbered and ranked according to appearance] and the formating of pages [restart numbering of pages in page 11], have not been corrected, although the authors stated that they did. Please correct.
Some comments
1. Page 4 line 46
The fact that the authors chose to do the research on literature on Rural planning in China, based only in scientific literature especially in Web of science journals, further aggravates the problem stated in my first review and not adequately answerec by the authors. Rural planning is based on policy documents, laws, decrees etc. and not on scientific literature. Hence, I am not sure someone can infer conclusions on the state of rural planning in China by counting the frequency of terms in scientific journals.
Three examples for that:
The authors state
Line 8-9 page 8
Triadic nature of rural development and prioritizing the economy
First of all, one would hardly call it a triadic nature since environment is not in the top 30 and social is the 22nd. It rather indicates an unbalanced discourse, biased one could argue, towards economy. But yet, we do not know what the actual programmatic documents and texts contain, so we cannot have a picture on rural planning in China.
Lines 34-35 page 3(?)
"Rural cultural space received attention in the perspective of anthropology, which shows the complexity and flexibility of local people’s interaction with the state in rural cultural governance(Chen 2016; Mei-Hui Yang 2004).
It is difficult to assess the interaction of local people with the state by the fact that there was reference to that in international journals.
Page 3(?) Lines 43 - 50
Participatory Assessment and Planning (PAP) and Participatory rural appraisal (PRA) approaches appear in literature as of 2002 ( Figure 5). These approaches enable all relevant actors to be involved in a ‘doing and learning’ process, in which the community can generate new knowledge, acquire new skills, build its capacity and confidence to interact with others, and innovate new approaches and technologies for its development (Ye, Wang, and Lu 2002). This envisages rural planning as a participatory process that empowers rural dwellers to take control of their own priorities for change (Ellis and Biggs 2001), along with emphasizing the uniqueness of local knowledge and individual experience, while rejecting overarching theories and large-scale programmes.
Yet, we do not know whether this approach has acquired any relevance in actual rural land planning procedures in China or not. Morever, according to the authors, participation is found in a rather inferior ranking in their research.
2. Page 4 footnote 3
The statements on the exclusion of certain features need more argumentation. For example
a. Should identification and support or mainstreaming of local bottom-up initiatives be excluded from rural planning?
b. The idea that rural planning is limited to co-ordinate given policies, programmes and projects leads to a totally top-down approach where the role of local "planners" is constrained to follow instructions from above and acommodate political agendas. And I have my doubts whether this is planning, to me it is mere bureaucratic administration.
FInally
3. I maintain my view that the very interesting part 5. Conclusion and discussion: the pursuit of development behind heterogeneous ideologies,
does not need the quantitative part of the manuscript. It stands alone based on the references to papers.
Author Response
We thank the reviewer for detailed and valuable comments. All of the suggestions made by the reviewer have been incorporated into the manuscript. Please see the revision letter in the attachment.

Reviewer 4 Report
The author gave a detailed response to my comments, and I have no other comments.
Author Response
We appreciate the reviewer for detailed suggestions, which are very helpful in modifying mistakes and improving the manuscript.